

# Determination of metabolite differences between loquat nectar and honey by UPLC-MS/MS

Xin Sun, Rui Shu, Yuli Qu and Junjun Dai

Sericultural Research Institute, Anhui Academy of Agricultural Sciences, Hefei, Anhui, China

## ABSTRACT

Honey is a widely consumed natural agricultural product. Honey bees produce honey by collecting nectar from different flowers and then metabolising it to form honey which is stored in the hive. The current focus of research has been on the primary characteristic metabolites of monofloral honey from different plant sources. There is a lack of understanding of the differences in the transformation and composition of overall metabolites between plant nectar and the honey that is processed by bees after nectar feeding. In this study, loquat nectar and mature loquat honey were used for the detection of all non-volatile metabolites in both by ultra performance liquid chromatography-tandem mass spectrometry (UPLC-MS/MS). Subsequently, an analysis was conducted on the primary metabolites, including saccharides, amino acids and their derivatives, nucleotides and their derivatives, lipids, and organic acids. In addition, the secondary metabolites, including flavonoids, phenolic acids, and terpenoid, were analysed. The results showed that there was a significant difference in the relative content of non-volatile metabolites between the two. We detected a total of 914 non-volatile metabolites, of which 834 were detected in loquat nectar and 759 in loquat honey. We analyzed the relative content of metabolites based on their classification and found that the relative content of primary and secondary metabolites showed different trends after processing by bees. Among them, the content of nucleotides and their derivatives and sugar metabolites in loquat honey was generally higher than that in loquat honey. This result presents a comprehensive picture of the non-volatile metabolite composition of loquat nectar and loquat honey, and systematically compares the changes in the relative content of the two substances.

## INTRODUCTION

Honey is a widely consumed, natural, premium food product that is considered high quality and valuable because of its unique flavor and texture (*Song et al., 2019*). Honey is collected by bees as nectar from flowers or insect secretions from plants, which are transformed, precipitated and dehydrated through the action of digestive enzymes and other biologically active substances and ultimately stored in the hive (*Da Silva et al., 2016*; *Power et al., 2018*; *Stern, Eisikowitch & Dag, 2001*). Compounds such as saccharidess, amino acids and phenolic acids, honey has recognized antioxidant and anti-inflammatory medicinal

Corresponding authors
Xin Sun, ahcsssx@163.com
Junjun Dai, 18855066300@163.com

properties, as well as many potential health benefits (*Bogdanov et al., 2008*; *Eteraf-Oskoue & Najafi, 2013*; *Naila et al., 2018*). Currently, researchers analyze the chemical composition of honey to determine its botanical origin as well as to tell if it is adulterated. *Smallfield, Joyce & Van Klink (2018)* investigated unique markers in Manuka honey and found that these markers were mainly produced during nectar production (*Smallfield, Joyce & Van Klink, 2018*). *Escriche et al. (2011)* finds unique markers for hesperidin and methyl anthranilate in *Citrus* honey (*Escriche et al., 2011*).

Nectar is one of the main resources for pollinating insects and is rich in sacchariess, amino acids and secondary metabolites (*Heil, 2011*). These chemicals not only affect the behavior of pollinating insects, but also play an important role in plant-pollinating insect interactions (*Adler & Irwin, 2012*; *Stevenson et al., 2016*). Sacchariess and amino acids in nectar are important basic food rewards in the reciprocal relationship between plants and pollinators, and different nectar properties affect bee collecting behavior and honey production potential (*Barberis et al., 2023*; *Bertazzini et al., 2010*; *Na, Kim & Park, 2024*; *Nicolson & Thornburg, 2007*). In recent years, researchers have begun to focus on low concentrations of substances in nectar, known as nectar secondary compounds (NSCs), including compounds such as non-protein amino acids and biogenic amines. More than mere nutrients, they regulate the behavior of pollinating insects and may even influence the nectar microbial community, thereby protecting the nectar from being used by other non-target species (*Bogo et al., 2019*; *Manson et al., 2013*; *Peng, Schroeder & Grüter, 2020*; *Stevenson, Nicolson & Wright, 2017*).

Nectar and honey are the raw materials and end products of bee production, and in recent years, researchers have begun to focus on the differences in metabolites between nectar and honey. The various enzymes and other bioactives that bees add when converting nectar into honey cause the difference between the two substances (*Nicolson, 2022*; *Yan et al., 2024*). In different clover varieties, honey bees influence the conversion of isoflavones in nectar and honey (*Sultana et al., 2024*). *Naef et al. (2005)* further revealed the complex role of bees on metabolites during honey processing by analyzing the volatile composition of linden nectar, bee stomach and mature honey (*Naef et al., 2005*). However, the identification of key secondary metabolites in Jarrah (*Eucalyptus marginata*) honey using high-performance thin layer chromatography (HPTLC) can be used to evaluate the characteristics of Jarrah honey (*Islam, Barbour & Locher, 2024*). It is regrettable that there are no extant studies that have conducted a comprehensive metabolomic analysis of target metabolites in nectar and honey.

Loquat (*Eriobotrya japonica* Lindl.), a fruit tree of the Rosaceae family, is not self-fertile and relies on xenogamy for fruit set, so insect activity is necessary for xenogamy during flowering (*Free, 1993*; *Goldway et al., 2015*; *Stern, Eisikowitch & Dag, 2001*). The pollen and nectar of loquat flowers were very attractive to pollinating insects, and bees were the main pollinating insects of loquat flowers (*McGregor, 1976*; *Nyska et al., 2014*). Researchers conducted a comprehensive study to examine the pharmacological effects of loquat leaves, flowers, fruits, seeds, and bark. The study's findings substantiated the presence of antioxidant, anti-inflammatory, anti-diabetic, anti-tumor, hypoglycemic, hepatoprotective and cardioprotective properties in loquat (*Lim, Yoo & Lee, 2023*). Research has identified

flavonoids and phenolic acids, the primary active substances in loquat, as possessing high antioxidant potential (*Syahputra et al., 2025*). Some terpenoid compounds have anti-inflammatory and anti-diabetic activity (*Banno et al., 2005*). Loquat honey has a unique flavor and cough suppressant properties. *Cheung et al. (2019)* identified hydroxybenzoic acid and protocatechuic aldehyde in loquat honey produced in China, and *Song et al.*'s (*2019*) study elucidated that a potential marker for loquat honey is chitinase (*Lin, Sharpe & Janick, 1999*).

In this study, we have used ultra performance liquid chromatography-tandem mass spectrometry (UPLC-MS/MS) for extensive analysis of targeted metabolites in loquat nectar, which is the raw material of the bee production process, and loquat honey, which is the end product, in order to reveal the differences of metabolites in loquat nectar and honey.

## MATERIALS AND METHODS

### Loquat nectar and honey samples

Raw floral nectar was collected from loquat flowers with disposable glass micro-capillaries in the morning during December 2023 in SheXian County, Anhui Province, China (29° 84′23″E; 118°06′45″N). On each of the three days, all flower nectar samples collected were pooled as individual samples (5–10 mL each), resulting in a total of three nectar samples. These were then filtered through 0.22 μm membrane filters (Merck Millipore) to remove pollen grains and dirt from the pooled nectar samples. We chose to conduct the experiment in a professional loquat orchard, randomly selecting Da Hongpao loquats that were growing uniformly. No other plants were grown in the orchard. To ensure that the loquat nectar and honey were not affected by other pollen, we physically isolated the loquat trees used for nectar collection with a 60-mesh insect net (Fig. S1A). In January 2023, three samples of loquat honey (1 kilogram each) were purchased from two trusted local beekeepers, whose hives were all less than 100 m from the nectar sampling site. These *Apis cerana cerana* were collected and produced honey from the beginning of the loquat flowering season in November 2022 until the end of the loquat flowering season in January 2023, with most of the collection taking place within insect-proof nets. The honey collected for the experiment was produced exclusively by *Apis cerana cerana.*

### Pollen testing of loquat nectar and honey

We conducted pollen analysis on the collected loquat flower nectar and honey. We weighed 25 g of loquat honey sample and dissolved it in 50 ml of distilled water. After thorough dissolution, we centrifuged the mixture at 2,500 rpm for 10 min and removed the supernatant. We repeated this step three times. We added acetic acid and sulfuric acid (1:9), boiled the mixture in a water bath for 5 min, cooled it to room temperature, washed the precipitate with distilled water, centrifuged it again, and then observed the pollen under a microscope. We took 100 μl of loquat honey, added 100 μl of distilled water, diluted the mixture, and observed it directly under a microscope. The results showed that the pollen detected in loquat honey was consistent with the pollen in loquat nectar (Figs. S1A and S1B). The authenticated honey was filtered twice, first through

an 80-mesh filter and then through a 400-mesh filter. Samples of the nectar and honey were stored at −80 °C for testing.

## Metabolomics analysis
### Sample preparation and extraction
All chemicals and reagents used in the experiment are listed in Table S1. The sample was extracted from the −80 °C refrigerator, thawed until there was no ice in the sample, vortexed for 10 s, and thoroughly mixed. A volume of 100 μL of the sample was transferred to the appropriate 1.5-mL centrifuge tube, accompanied by the addition of 100 μL of 70% methanol with internal standard extraction solution (less than 100 μL, with the extraction solution applied at a volume ratio of 1:1 [V/V]). The samples were vortexed for 15 min at 12,000 revolutions per minute (r/min) at 4 °C, followed by a 3-minute centrifugation. The upper layer was extracted, filtered through a microporous membrane (0.22-micrometer pore size), and stored in the injection vial for liquid chromatography-tandem mass spectrometry (LC-MS/MS) detection.

### UPLC conditions
The sample extracts were analyzed using an UPLC-ESI-MS/MS system (UPLC, ExionLC™ AD, https://sciex.com.cn/) and Tandem mass spectrometry system (https://sciex.com.cn/). The analytical conditions were as follows, UPLC: column, Agilent SB-C18 (1.8 μm, 2.1 mm * 100 mm); The mobile phase was consisted of solvent A, pure water with 0.1% formic acid, and solvent B, acetonitrile with 0.1% formic acid. Sample measurements were performed with a gradient program that employed the starting conditions of 95% A, 5% B. Within 9 min, a linear gradient to 5% A, 95% B was programmed, and a composition of 5% A, 95% B was kept for 1 min. Subsequently, a composition of 95% A, 5.0% B was adjusted within 1.1 min and kept for 2.9 min. The flow velocity was set as 0.35 mL per minute; The column oven was set to 40 °C; The injection volume was 2 μL. The effluent was alternatively connected to an ESI-QTOF-MS or ESI-triple quadrupole-linear ion trap (QTRAP)-MS.

### ESI-Q TOF 6600-MS/MS
The data acquisition was operated using the information-dependent acquisition (IDA) mode. The source parameters were set as follows: ion source gas 1 (GAS1), 50 psi; ion source gas 2 (GAS2), 50 psi; curtain gas (CUR), 25 psi; temperature (TEM), 500 °C; declustering potential (DP), 60 V, or −60 V in positive or negative modes, respectively; and ion spray voltagefloating (ISVF), 5,500 or −4,500 V in positive or negative modes, respectively. The TOF MS scan parameters were set as follows: mass range, 100–1,250 Da; accumulation time, 200 ms; and dynamic background subtract, on. The product ion scan parameters were set as follows: mass range, 50–1,250 Da; accumulation time, 50 ms; collision energy, 30 or −30 V in positive or negative modes, respectively; collision energy spread, 15; resolution, UNIT; charge state, 1 to 1; intensity, 100 cps; exclude isotopes within 4 Da; mass tolerance, 50 ppm; maximum number of candidate ions to monitor per cycle, 10; and exclude former target ions, for 4 s after two occurrences.

### ESI-Q TRAP-MS/MS

The ESI source operation parameters were as follows: source temperature 550 °C; ion spray voltage (IS) 5,500 V (positive ion mode)/−4,500 V (negative ion mode); ion source gas I (GSI), gas II(GSII), curtain gas (CUR) were set at 50, 60, and 25 psi, respectively; the collision-activated dissociation (CAD) was high. QQQ scans were acquired as MRM experiments with collision gas (nitrogen) set to medium. DP (declustering potential) and CE (collision energy) for individual MRM transitions was done with further DP and CE optimization. A specific set of MRM transitions were monitored for each period according to the metabolites eluted within this period.

### Qualitative and quantitative analyses of metabolites

High resolution mass spectrometry AB sciex TripleTOF6600 was used for qualitative detection of positive and negative mixed samples, and then AB sciex 4500 QTRAP was used for quantitative detection of each sample. Combining the advantages of non-targeted and targeted metabolomics, high-resolution mass spectrometry was used for accurate qualitative analysis, and triple quadrupole mass spectrometry with high sensitivity, high specificity, and excellent quantitative ability was used as a supplementary tool. The parameters matched in metabolite identification include Q1 precise molecular weight, secondary fragments, retention time, and isotopes. The qualitative and quantitative mass spectrometry analysis of metabolites in project samples is based on the MetWare database (MWDB) and multi reaction monitoring (MRM). Metabolite identification is based on the precise mass of metabolites, MS2 fragments, MS2 fragment isotope distribution, and retention time (RT). Through our company's independently developed intelligent secondary spectrum matching method, the secondary spectrum and RT of metabolites in project samples are intelligently matched with the company's database secondary spectrum and RT one by one. The MS tolerance and MS2 tolerance are set to 20 ppm, and the RT tolerance is 0.2 min. The standard sample information used for quantitative analysis in the experiment is shown in Table S2. Compounds with relatively large differences in relative content or compounds that have a common impact on human health were selected for analysis. The RT data for these compounds are shown in Table S3.

## Statistical analysis
### PCA

Unsupervised principal component analysis (PCA) was performed by statistics function prcomp within R (base package 4.1.2) (http://www.r-project.org). The data was unit variance scaled before unsupervised PCA.

### Hierarchical cluster analysis and pearson correlation coefficients

The hierarchical cluster analysis (HCA) results of samples and metabolites were presented as heatmaps with dendrograms, while Pearson correlation coefficients (PCC) between samples were caculated by the cor function in R (base package 4.1.2) and presented as only heatmaps. Both HCA and PCC were carried out by R package ComplexHeatmap. For HCA, normalized signal intensities of metabolites (unit variance scaling) are visualized as a color spectrum.

*Differential metabolites selected*

For two-group analysis, differential metabolites were determined by Variable Importance in Projection (VIP, VIP > 1) and absolute $Log_2$(Fold Change) ($Log_2FC$, $|Log_2FC| \geq 1.0$). VIP values were extracted from orthogonal partial least squares-discriminant analysis (OPLS-DA) result, which also contain score plots and permutation plots, was generated using R package MetaboAnalystR (1.0.1). The data was log transform ($log_2$) and mean centering before OPLS-DA. In order to avoid overfitting, a permutation test (200 permutations) was performed.

## RESULTS

### Whole metabolome-scale comparative analysis of loquat nectar and honey

In order to examine the diversity of metabolites, loquat nectar and loquat honey, namely DHP-HM and DHP-FM, were selected to perform a metabolomic analysis. Then, UPLC-MS-based widely targeted metabolomics was carried out, in which 914 metabolites were detected (Table S4). As shown in Fig. 1A, the 914 metabolites can be classified in detail according to their properties: 130 flavonoids, 127 phenolic acids, 118 amino acids and derivatives, 100 alkaloids, 97 lipids, 72 organic acids, 50 nucleotides and derivatives, 46 saccharides, and 42 terpenes. phenolics, lignans, coumarins, vitamins, aldehydes, ketones, alcohols, quinones, chromones, lactones, steroids, tannins, and 35 other compounds.

Quality control (QC) samples, composed of a mixture of 50 µL of each sample, in triplicate, were analyzed to assess the reliability of the method. As demonstrated in Figs. 1B and 1C, the overlap analysis of the total ion current diagram of the QC sample in positive and negative ionization modes exhibited satisfactory repeatability. Utilizing the comprehensive non-volatile metabolite dataset (914 metabolites) obtained in this study, a non-supervised PCA model was implemented for further analysis of the QC sample and the tested sample (Fig. 1D).

To further explore the diversity of nutrients between DHPHM and DHPFM, we analyzed metabolic profiling using PCA and HCA. According to the PCA, two principal components, PC1 and PC2, represented a 78.21% and 8.09% contribution to the differences among DHPHM and DHPFM, respectively (Fig. 1D). This outcome serves to further substantiate the validity of the employed methodology. Furthermore, the results indicated that the non-volatile metabolites present in DHPHM and DHPFM were distinct.

### Significantly different non-volatile metabolites between loquat nectar and honey

A total of 834 non-volatile metabolites were detected in DHPHM and 759 in DHPFM by UPLC-MS-MS method. The metabolites were then ranked according to their detection frequency. The top five non-volatile metabolites detected in DHPHM were phenolic acids, flavonoids, amino acids and their derivatives, alkaloids, and lipids. In DHPFM, the top five were amino acids and their derivatives, flavonoids, lipids, alkaloids, and phenolic acids, in that order. This finding indicates that the nonvolatile metabolite compositions of loquat nectar and loquat honey are distinct (Fig. S2).
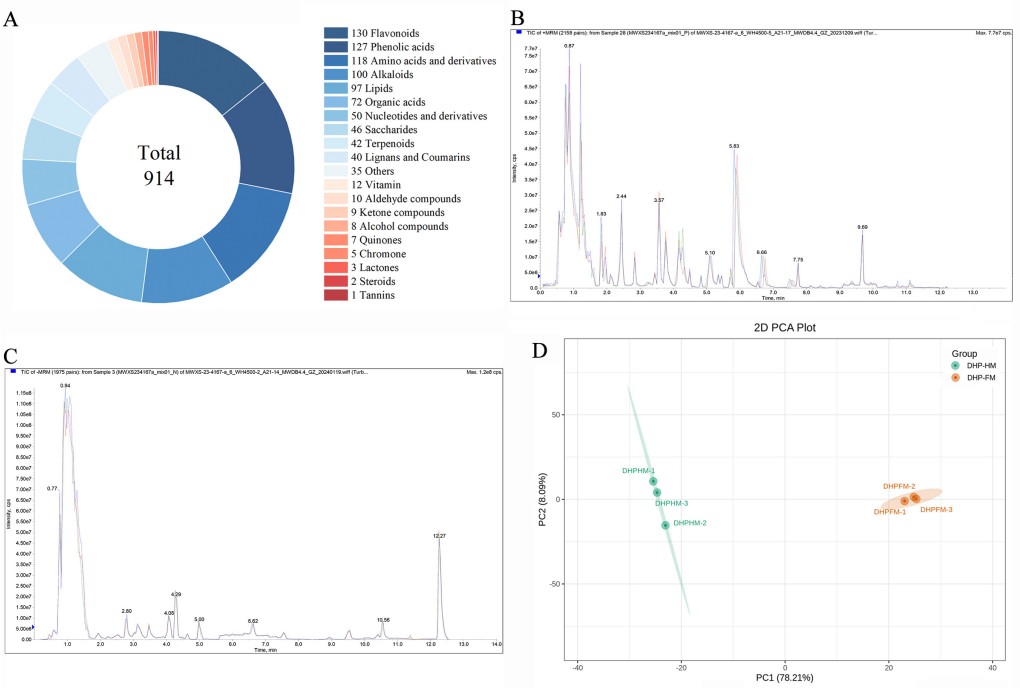

**Figure 1** **Non-volatile metabolite classes in DHPHM and DHPFM obtained by a wide range of targeted metabolism methods and assessment of method reliability.** (A) DHPHM and DHPFM are respectively the nectar of the Da Hongpao loquat variety and the honey produced by *Apis cerana cerana* from the Da Hongpao loquat. (B) Overlay analysis of total ion current maps of quality control (QC) samples in positive ion mode. (C) Overlap analysis of total ion current maps of quality control (QC) samples in negative ion mode. (D) PCA score plots for all samples tested.

OPLS-DA was performed to ascertain significant differences between non-volatile metabolites of DHPHM and DHPFM. VIP>1.0 and FC≥2/≤0.5 were used to indicate significant differences in non-volatile metabolites. As demonstrated in Fig. 2A and Table S5, a total of 595 significantly different non-volatile metabolites were identified, including 105 flavonoids, 99 phenolic acids, 85 amino acids and derivatives, 58 alkaloids, 43 lipids, 43 organic acids, 31 nucleotides and derivatives, 30 lignans and coumarins, 26 terpenoids, 20 saccharides, 10 vitamins, six ketone compounds, six quinones, four alcohol compounds, two chromones, two steroids, one tannins, and 24 others. To further elucidate the significant changes in these non-volatile metabolites, heatmaps of some categories are illustrated in Fig. 2B, and the detailed results are presented below.

### Flavonoids

In a recent study, a total of 130 non-volatile metabolites of flavonoids were detected in DHPHM and DHPFM. These include flavonoids, flavonols, anthocyanins, flavanones, and some flavanols. During the conversion of DHPHM and DHPFM, the levels of 94 flavonoids decreased, while the levels of 11 increased (Fig. 3A). Specifically, 29 flavonoids were identified in DHPHM, while seven were detected in DHPFM (Fig. 3B and Table S6). Subsequent analysis revealed a substantial decrease in the relative content of quercetin,
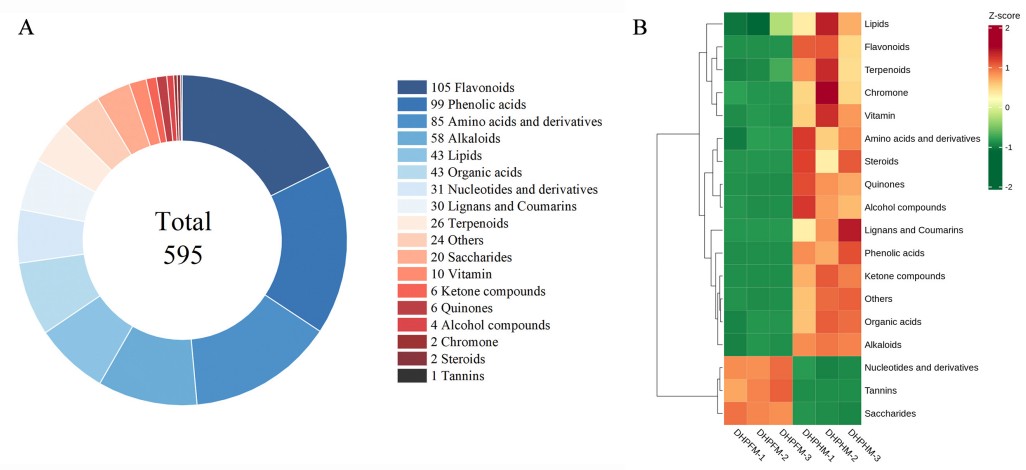

**Figure 2** **Comparison of differential metabolites between DHPHM and DHPFM.** (A) Classification of significantly different non-volatile metabolites in DHPHM and DHPFM. (B) Heatmap of significantly different non-volatile metabolite content in DHPHM and DHPFM. Metabolite content data were processed using UV (unit variance scaling) and plotted as heat maps using the R complexHeatmap package. Different colors represent different values obtained after standardization of different relative contents (red represents high content, green represents low content).

rutin, hesperidin, catechin, and cynaroside in DHPFM, while a significant increase in the relative content of morin was observed (Fig. 3C). These findings underscore the reliability of flavonoids as a means of traceability in honey analysis and highlight the significant role of bee metabolism in the transformation of flavonoids during honey production.

### Phenolic acids

As illustrated in Fig. 4A, a total of 127 phenolic acid nonvolatile metabolites were detected in DHPHM and DHPFM, second only to the number of flavonoids. Subsequent analysis revealed the absence of four phenolic acid compounds (2-Phenylethanol, Bengenin, Cornoside, and Demethyl coniferin) in loquat nectar. Furthermore, a comparison of the two samples demonstrated that 42 phenolic acid compounds were not detected in DHPFM (Fig. 4B and Table S6). Subsequent analysis of the prevalent phenolic acid compounds in both DHPHM and DHPFM revealed a decline in their relative abundance in honey (Fig. 4C). The findings indicated that loquat nectar is abundant in phenolic acids with high relative levels, which may be more beneficial to the plant in attracting specific pollinating insects. Additionally, some flavor-enhancing phenolic acids are present in loquat honey processed by honey bees.

### Amino acids and derivatives

As illustrated in Fig. 5A, a total of 118 non-volatile metabolites of amino acids and their derivatives were detected in DHPHM and DHPFM. A comparative analysis revealed a decline in 65 amino acids and their derivatives, while 20 exhibited an increase in DHPFM compared to DHPHM. Our analysis identified 11 amino acids and their derivatives in DHPFM that were not detected in DHPHM, while 15 amino acids and their derivatives

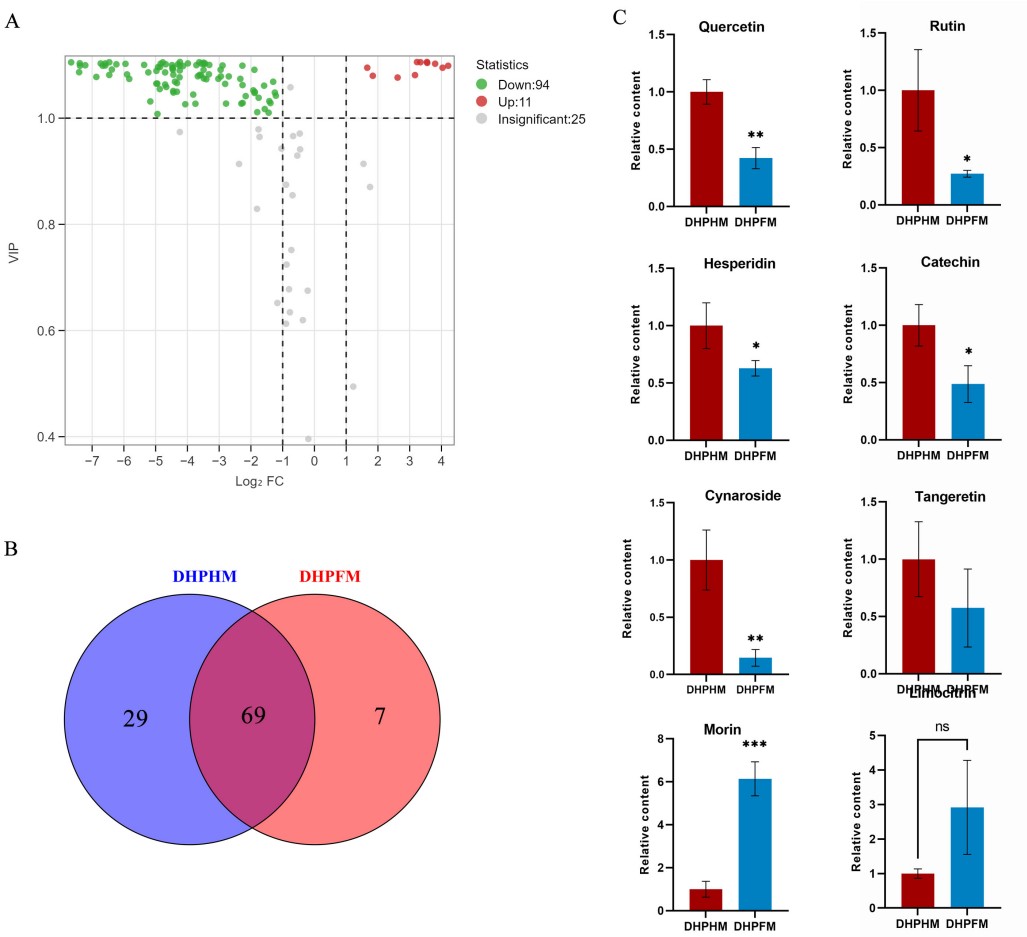

**Figure 3 Comparison of flavonoid differential metabolites.** (A) Volcano plot of the number of changes in relative flavonoid metabolite content in DHPHM compared to DHPFM. (B) Number of individual flavonoid metabolites detected in DHPHM and DHPFM. (C) Comparison of relative flavonoid metabolite content. Data are expressed as mean $\pm$ standard deviation ($n = 3$ biological replicates). Statistical significance between different samples was assessed using a $t$-test (* indicates $P < 0.05$, ** indicates $P < 0.01$, ***indicates $P < 0.001$).

were detected in DHPHM. It was observed that a greater number of amino acid derivatives were present in nectar, while in honey, they were present as amino acids. This observation suggests that the process of plant secondary metabolism is more active in the honey sample, while the substances present in the nectar are more stable (Fig. 5B and Table S6). The relative contents of amino acids and their derivatives common to both were analyzed, and the results showed that the contents of the nonprotein amino acids L-citrulline and L-ornithine in DHPFM were significantly lower than those in DHPHM. Among the protein amino acids, the contents of the aromatic amino acid L-tryptophan were significantly lower, whereas the contents of the aliphatic amino acids L-lysine and L-glutamine were significantly higher (Fig. 5C).
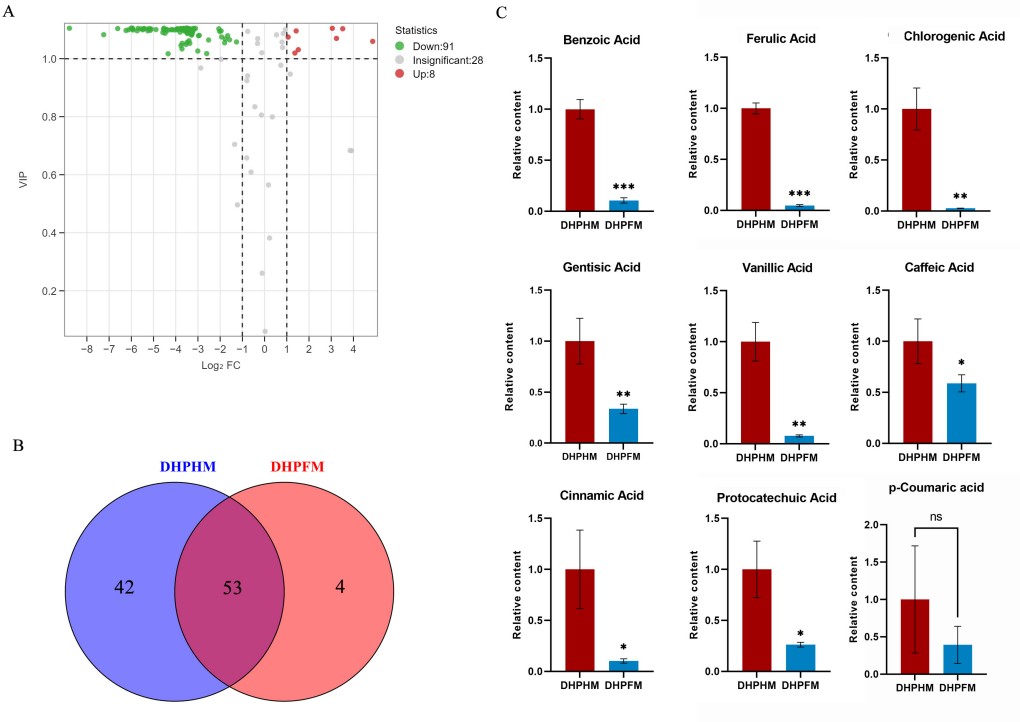

**Figure 4** **Comparison of phenolic acids differential metabolites.** (A) Volcano plot of the number of changes in relative phenolic acids metabolite content in DHPHM compared to DHPFM. (B) Number of individual phenolic acids metabolites detected in DHPHM and DHPFM. (C) Comparison of relative phenolic acids metabolite content. Data are expressed as mean $\pm$ standard deviation ($n = 3$ biological replicates). Statistical significance between different samples was assessed using a $t$-test (* indicates $P < 0.05$, ** indicates $P < 0.01$, ***indicates $P < 0.001$).

## Saccharides

A comprehensive analysis of DHPHM and DHPFM revealed the presence of 46 saccharides nonvolatile metabolites. Of particular note, 3-phospho-D-glyceric acid and L-Arabitol were exclusively detected in DHPHM, while five distinct saccharide metabolites, including D-Panose, were identified in DHPFM (Fig. 6B). Of the 39 saccharides metabolites detected in both, the relative levels of eight saccharides were found to be significantly up-regulated in DHPFM, and the relative levels of five saccharides were found to be significantly higher in DHPHM (Fig. 6A). Subsequent analysis of select saccharides revealed that the D-glucose content of honey was significantly higher than that of nectar, while D-fructose exhibited no significant change during the transformation process (Fig. 6B). The gradual consumption of glucan, maltopentaose, and D-ribose during honey transformation was observed, while raffinose and D-mannose exhibited a corresponding gradual accumulation (Fig. 6C). The results demonstrated that the saccharides metabolites in DHPFM and DHPHM exhibited significant differences, and the saccharides metabolites in loquat honey processed by honey bees were more abundant than those in loquat nectar, with a substantial increase in monosaccharide content.

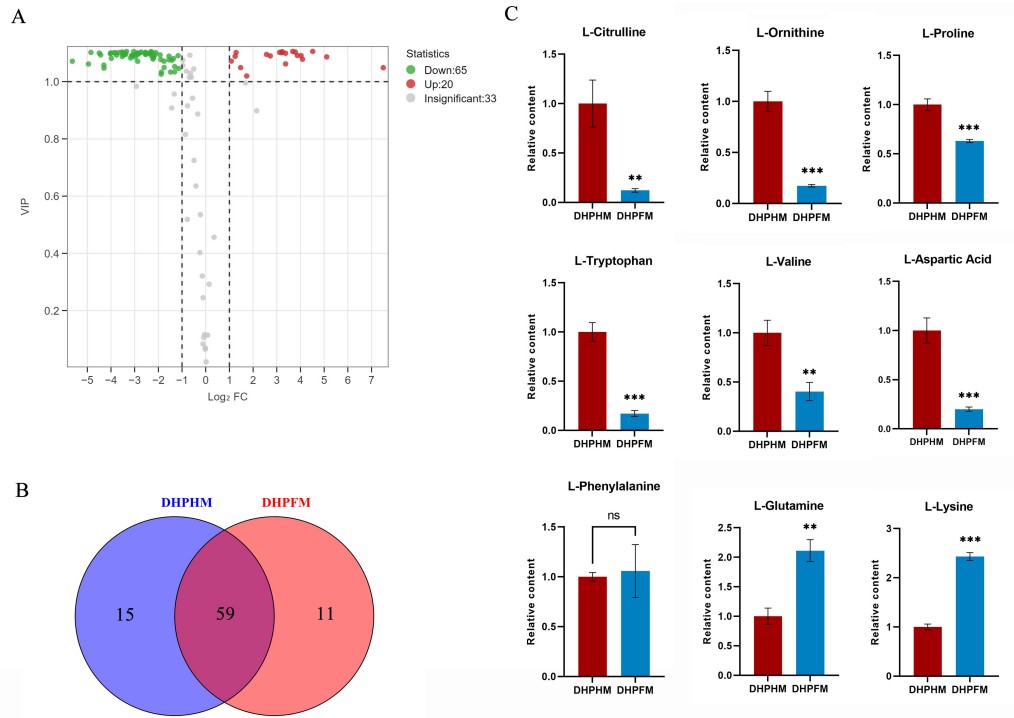

**Figure 5 Comparison of amino acids and derivatives differential metabolites.** (A) Volcano plot of the number of changes in relative amino acids and derivatives metabolite content in DHPHM compared to DHPFM. (B) Number of individual amino acids and derivatives metabolites detected in DHPHM and DHPFM. (C) Comparison of relative amino acids and derivatives metabolite content. Data are expressed as mean ± standard deviation ($n = 3$ biological replicates). Statistical significance between different samples was assessed using a $t$-test (* indicates $P < 0.05$, ** indicates $P < 0.01$, *** indicates $P < 0.001$).

## Comparative analysis of other metabolites

A comprehensive evaluation of the metabolite distinctions between DHPHM and DHPFM was conducted through a subsequent analysis of the alterations in the composition of organic acids, terpenoids, lipids, nucleotides, and their derivatives in DHPHM and DHPFM (Fig. 7).

Despite comprising less than 0.5% of the honey's composition, organic acids play a significant role in shaping its organoleptic, physical, and chemical characteristics (*Mato et al., 2003*). A comprehensive analysis of DHPHM and DHPFM revealed the presence of a total of 72 organic acids. As illustrated in Fig. 7A, DHPHM exhibited a higher cumulative content of 4-Guanidinobutyric acid, Adipic Acid, Suberic Acid, 2-Methylglutaric acid, Citric acid-1-O-diglucoside, 2-Acetyl-2-Hydroxybutanoic Acid, and Tropic acid. Conversely, 6-Hydroxyhexanoic acid, 2-Aminoisobutyric acid, Tianshic acid, and 2-Picolinic acid were found to be more abundant in DHPFM. The findings indicated that the composition and relative content of organic acids in loquat nectar and honey were distinct.

A total of 42 terpenoids were detected, and further analysis of their differential metabolites revealed a significantly higher relative content of 6′-O-caffeoylharpagide, eucommioside, and hydroxycarvine in DHPFM compared to DHPHM. A notable increase

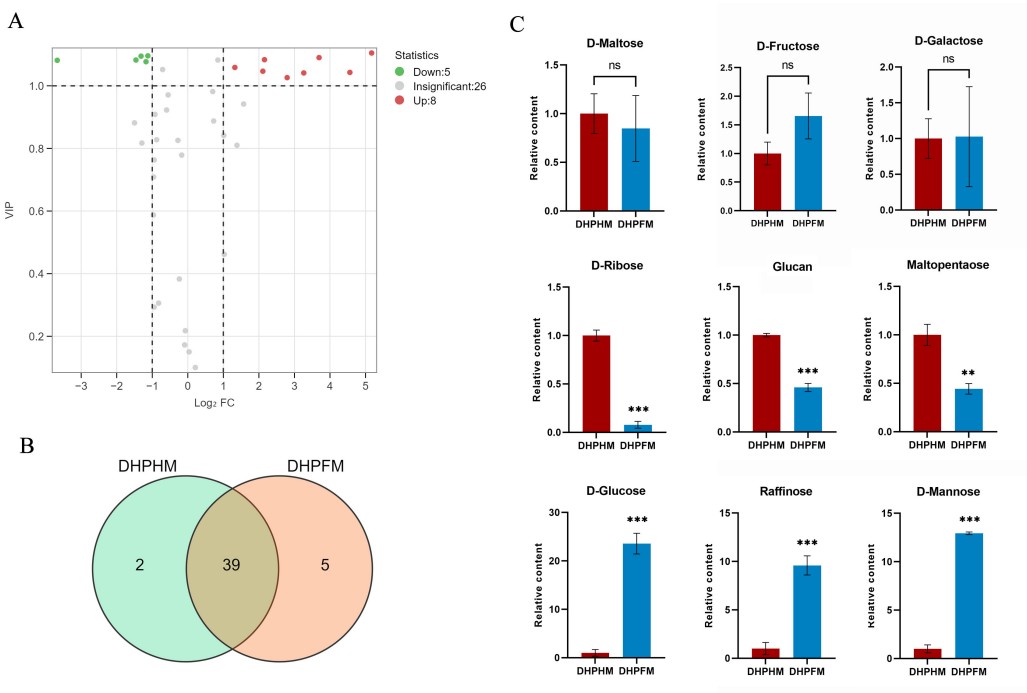

**Figure 6 Comparison of saccharides differential metabolites.** (A) Volcano plot of the number of changes in relative saccharides metabolite content in DHPHM compared to DHPFM. (B) Number of individual saccharides metabolites detected in DHPHM and DHPFM. (C) Comparison of relative saccharides metabolite content. Data are expressed as mean ± standard deviation ($n = 3$ biological replicates). Statistical significance between different samples was assessed using a $t$-test (* indicates $P < 0.05$, ** indicates $P < 0.01$, ***indicates $P < 0.001$).

in the relative content of niga-ichigoside F3, dihydrophaseic acid, and trachelosperonide B-1 was observed in DHPHM (Fig. 7B).

A total of 97 lipid metabolites were detected in DHPHM and DHPFM, 23 of which were exclusively present in DHPFM. The majority of these detections were lysophosphatidylethanolamine (LPE). Additionally, three free fatty acids were exclusively detected in DHPHM. A total of 23 differential metabolites were identified as shared by both, and the relative levels of all lipid metabolites were found to be significantly higher in DHPFM compared to DHPHM (Fig. 7C). These findings imply that the accumulation of lipid metabolite content plays a pivotal role in the formation of loquat honey.

A total of 50 nucleotides and their derivatives were detected in DHPHM and DHPFM. A more balanced number of nucleotides and their derivatives with different relative contents was found in DHPHM and DHPFM (Fig. 7D). The levels of 5′-monophosphate, cytidine, 9-arabinosyladenine, and Adenosine exhibited a gradual accumulation trend in DHPFM. Conversely, the levels of 2′-deoxyguanosine, thymidine, thymine, and cytarabine were found to be higher in DHPHM.

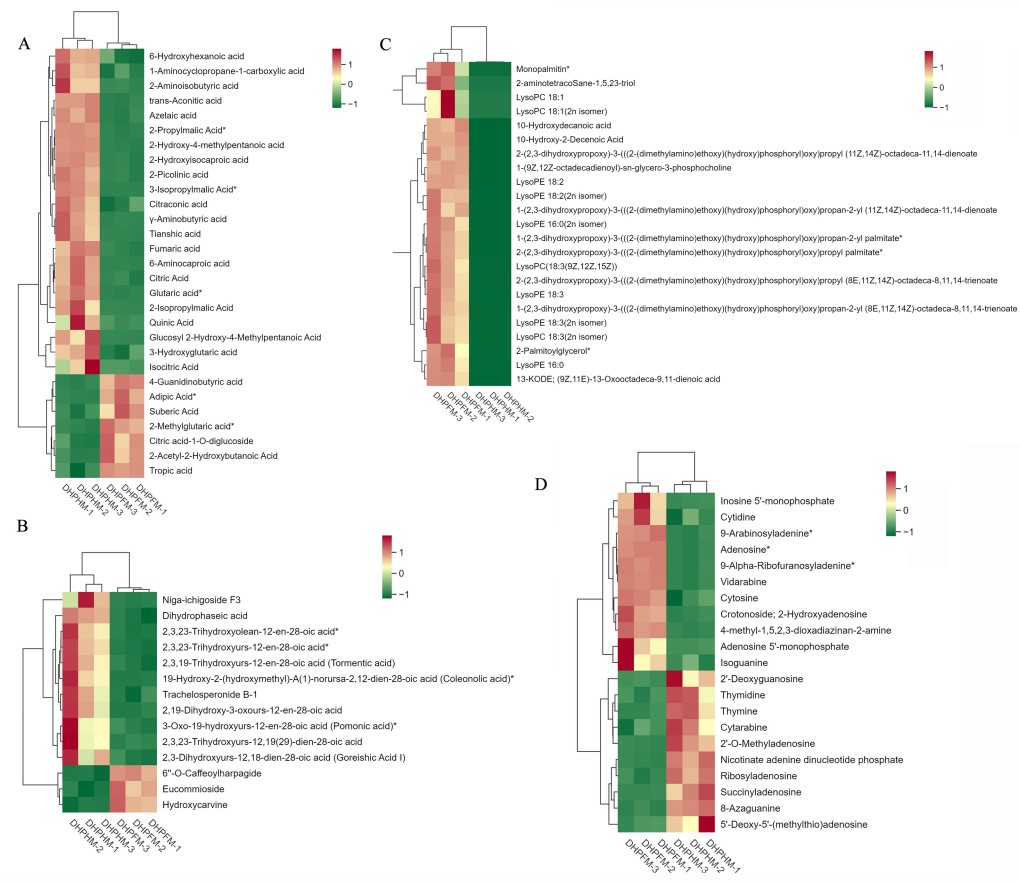

**Figure 7** **Accumulation of other partial metabolites in DHPHM and DHPFM.** (A) Heat map of organic acids in DHPHM and DHPFM. (B) Thermogram of terpenoids in DHPHM and DHPFM. (C) Thermogram of lipids in DHPHM and DHPFM. (D) Heat map of nucleotides and their derivatives in DHPHM and DHPFM. Metabolite content data were processed using UV (unit variance scaling) and plotted as heat maps using the R complexHeatmap package. Different colors represent different values obtained after standardization of different relative contents (red represents high content, green represents low content).

# DISCUSSION

We selected loquat as a subject for our study, focusing on the differences in metabolites between its nectar and honey. This decision was influenced by the distinctive characteristics of loquat, which blooms during the winter season. The colder weather during this time reduces the activity rate of various pollinating insects, thereby increasing the honey bee's specialization in collecting loquat nectar. *Stevenson, Nicolson & Wright*'s research (*2017*) revealed that gamma-aminobutyric acid (GABA) in nectar is attractive to a wide range of insects, including long-tongued bees, andrenid bees, and anthophorid bees (*Stevenson, Nicolson & Wright, 2017*). Conversely, asparagine has been found to repel insects such as bugs, beetles, wasps, anthomyiid flies, wasps, and short-tongued bees. Our research, which included the analysis of loquat nectar, detected asparagine but not GABA, thereby further corroborating the symbiotic relationship between winter bees and loquat flowers.

Conversely, the majority of prior studies have centered on the identification of individual compounds in nectar or honey (*Biller et al., 2015*; *Manson, Otterstatter & Thomson, 2010*). A substantial body of research has demonstrated that pollinating insects play a crucial role in the natural control of pathogens by consuming specific secondary metabolites present in nectar. This process contributes to the reduction of pathogen proliferation and dissemination. For instance, the alkaloid pseudopyrrolizidine alkaloid present in tobacco plant nectar has been observed to reduce the load of pathogens in bumblebees (*Bombus terrestris*) and to impede the dissemination of the parasitic unicellular eukaryote (*Crithidia bombi*) (*Anthony et al., 2015*; *Richardson et al., 2015*). Research on honey metabolites has primarily focused on techniques for determining honey sources using specific compounds or identifying individual metabolic compounds. For instance, the most prevalent metabolites in buckwheat honey encompass aldehydes, such as 3-methylbutanal, 2-methylbutanal, and 3-methylbutyric acid (*Panseri et al., 2013*; *Wolski et al., 2006*). In contrast, *Biesaga & Pyrzynska (2009)* employed HPLC-ESI-MS/MS to ascertain phenolic acid compounds in honey.

The differential metabolite profile between the two groups is primarily attributable to the consumption of loquat nectar by bees, which is essential for their growth and development. Additionally, the processing and storage of honey contributes to the variation in metabolite profiles. The most abundant metabolites in loquat nectar and honey are flavonoids and phenolic acids. The phenolic compounds present in nectar have been shown to possess an attractant effect, thereby attracting effective pollinators and enhancing their loyalty to the plant (*Zhang et al., 2018*). The flavonoids and phenolic acids present in honey exhibit significant antioxidant properties, which play a crucial role in the analysis of the floral and geographical origins of honey (*Yao et al., 2004*). Flavonoids and phenolic acids have also been demonstrated to assist bees in reducing the toxicity of pesticides and in resisting disease infections. For instance, bees can utilize quercetin, found in nectar or pollen, to mitigate the toxicity of the acaricides tau-fluvalinate, thereby extending their lifespan (*Mao, Schuler & Berenbaum, 2011*; *Riddick, 2021*). The findings indicate a general decline in the relative abundance of phenolic acids and flavonoids during the transformation of loquat nectar into loquat honey through the involvement of bees. This observation substantiates the ingestion of flavonoids and phenolic acids during the growth and development phase of the bee from the perspective of substance metabolism.

Amino acids and their derivatives in nectar play a pivotal role in the growth and development of bees. They contribute significantly to the formation of nectar's unique flavor and serve as an important nitrogen source for bees (*Nicolson, 2022*; *Roy et al., 2017*). In contrast, the proteins and amino acids present in honey are of twofold origin: they are derived from the bees themselves as well as from the nectar and pollen of plants (*Hermosín, Chicón & Dolores Cabezudo, 2003*). Moreover, they constitute a predominant category of trace metabolites derived from plant sources. L-lysine and DL-methionine have been identified as critical factors in the hierarchical differentiation of bee larvae (*Chen et al., 2021*). Our research findings revealed the presence of DL-methionine exclusively in loquat nectar, while L-lysine exhibited significantly higher relative levels in loquat honey. This observation is attributed to the enhanced stability and suitability of loquat honey for

storage within the hive, thereby facilitating bee growth and development. Furthermore, we observed elevated relative levels of L-proline in both loquat nectar and honey. It has been documented that low-temperature stress prompts loquat flowers to accumulate higher levels of proline, while bees require greater proline support for winter flight (*Carter et al., 2006*; *Teulier et al., 2016*). This finding underscores the necessity for further investigation into the mechanisms by which bees process nectar into honey. Furthermore, the analysis of diverse metabolites led to the identification of specific substances exclusively detected in loquat nectar or honey. These findings necessitate further investigation to ascertain their precise metabolic pathways and functional roles.

The predominant nutrients found in nectar are sucrose and its derivatives, including hexose, glucose, and fructose (*Nicolson, 2022*). A variety of factors have been demonstrated to influence bee behavior and pollination efficiency, including the sucrose-hexose ratio, sugar concentration and composition, and nectar secretion efficiency in nectar (*Baude et al., 2016*; *Herrera, Pérez & Alonso, 2006*). It has been demonstrated that bees exhibit a higher degree of efficiency in the digestion of sucrose and glucose in comparison to other types of sugar. Sucrose and glucose have been demonstrated to prolong the lifespan of bees, while high fructose concentrations have been shown to induce digestive system disorders in bees (*Abdella et al., 2024*; *Ledesma González et al., 2025*). Among them, worker bees secrete $\alpha$-glucosidase from their hypopharyngeal glands to convert sucrose, the main component of nectar, into glucose and fructose (*Kunieda et al., 2006*). The results of our study demonstrate that the content of polysaccharides, including maltose and glucan, in nectar undergoes a substantial decrease following decomposition by bees. Concurrently, the glucose content in honey experiences a significant increase. The results of this study confirm the process by which bees use their own secreted enzymes to convert nectar into honey. Furthermore, a substantial increase in the relative content of cotton sugar and mannitol was observed in loquat honey. It is hypothesized that bees may have collected some of the honeydew secreted by other parts of the loquat tree while collecting nectar.

An examination of the metabolite fingerprints of loquat nectar and honey was conducted using UPLC-MS/MS, with a focus on elucidating the significant differences in metabolites between these two samples. However, the present study is not without its limitations, primarily due to an inability to fully demonstrate the complete pathway by which individual metabolites are transformed from source nectar into mature honey through bee processing. This represents a significant gap in the research on bee honey processing. In the subsequent stage of the study, we will undertake a more precise decomposition and analysis of the metabolite differences present at each stage of the honey-making process. This will include the sucking, processing, and fermentation of honey by bees. Our objective is to achieve a precise identification of the synthetic metabolic pathways of metabolites that are beneficial to human health. This will facilitate a more profound comprehension of the metabolic processes involved in bee honey production and the compounds present in it. Notwithstanding, the findings of this study underscore the reliability of UPLC-MS/MS technology in the realm of metabolite detection. The results accurately revealed the complete metabolome of loquat nectar and loquat honey, analyzing the major types of compounds

within them. This will deepen our understanding of the metabolome of monofloral honey and enhance the metabolomics data for monofloral honey and plant nectar.

## CONCLUSION

A broad-targeted metabolomics approach, based on UPLC-MS/MS, was employed to detect differences in the composition of metabolites in loquat nectar and loquat honey. This approach was used to explain the biological significance of bees' involvement in the honey-making process as producers. The study's findings revealed significant disparities in the chemical constituents of loquat nectar and loquat honey, which play a crucial role in the development of the distinctive characteristics of loquat honey, including its color, flavor, and antimicrobial and anti-inflammatory properties. The study further elucidated the biological functions and significance of non-volatile metabolites in the honeybee's involvement in the conversion of loquat nectar into loquat honey. Additionally, potential targets for the identification of non-volatile metabolites in loquat honey were proposed.

## ACKNOWLEDGEMENTS

The honey samples were provided by the Association of Chinese Bee Breeding Base in Shendu Town, Shexian County. We thank Dr. Xueqiang Su and Dr. Feng Chen for their help in the research.

### Funding
This research was supported by Open Project of Anhui Key Laboratory of Resource Insect Biology and Innovative Utilization (FKLRIB202404); Anhui Characteristic Agricultural Industry Technology System (Anhui Agricultural Science and Technology [2021] No. 711); Anhui Academy of Agricultural Sciences Youth Fund Project (2025YL024); Anhui Academy of Agricultural Sciences Achievement Transformation Project (2025ZH011); Anhui Academy of Agricultural Sciences Talent Project (XJBS-202446). The funders had no role in study design, data collection and analysis, decision to publish, or preparation of the manuscript.

### Grant Disclosures
The following grant information was disclosed by the authors:
Open Project of Anhui Key Laboratory of Resource Insect Biology and Innovative Utilization: FKLRIB202404.
Anhui Characteristic Agricultural Industry Technology System (Anhui Agricultural Science and Technology [2021]): 711.
Anhui Academy of Agricultural Sciences Youth Fund Project: 2025YL024.
Anhui Academy of Agricultural Sciences Achievement Transformation Project: 2025ZH011.
Anhui Academy of Agricultural Sciences Talent Project: XJBS-202446.

## Competing Interests

The authors declare there are no competing interests.

## Author Contributions

- Xin Sun conceived and designed the experiments, performed the experiments, analyzed the data, prepared figures and/or tables, and approved the final draft.
- Rui Shu conceived and designed the experiments, performed the experiments, prepared figures and/or tables, and approved the final draft.
- Yuli Qu conceived and designed the experiments, analyzed the data, authored or reviewed drafts of the article, and approved the final draft.
- Junjun Dai conceived and designed the experiments, authored or reviewed drafts of the article, and approved the final draft.

## Data Availability

The data is available at MetaboLights: MTBLS12228; and Zenodo: sun, . xin ., shu, . rui ., qu, . yuli ., & dai, . junjun . (2025). Determination of metabolite differences between loquat nectar and honey by UPLC-MS/MS [Data set]. Zenodo. Available at https://doi.org/10.5281/zenodo.15846598.

## Supplemental Information

Supplemental information for this article can be found online at http://dx.doi.org/10.7717/peerj.19988#supplemental-information.

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
