# Peer review of "Determination of metabolite differences between loquat nectar and honey by UPLC-MS/MS"

_PeerJ, doi:10.7717/peerj.19988_

## Round 0.1 · original submission · Major Revisions

General comments:

In this manuscript, Sun et al. used mass spectrometry to compare the content of loquat honey vs. loquat nectar. The goal of the study is to understanding the content of loquat honey and nectar, which has implications for human health.

There are two major weaknesses of this study. Reviewer #1 identified a critical weakness in study design: since the bees were not in a controlled environment, they honey they produced could have been from plants other than loquat. Since the main goal of this study is to identify metabolites in loquat honey vs. loquat nectar, the entire study and its conclusions may be flawed.

Along with Reviewer #1, I also believe there is a lot of missing information on the experimental methods, definitions in figure captions, etc. that hinder understanding the study. Also, the authors should realize PeerJ is not a metabolomics journal, so some brief statements need to be added throughout.

Specific comments:

Define "DHPHM" and "DHPFM". This was a major omission. Also, what are the numbers? Are these different samples? This is very confusing.

The reproducibility was shown on biplot of PCA1 and PCA2. Similar to the above comment: are these different samples or the same one analyzed 3 times? This is very confusing. Clearly explain technical replicates vs. independent samples.

Figure 1B: since the heatmap is already here, perform HCA and include the dendrogram, which can provide interesting insights. It will make the analysis consistent with Figure 7 and make the analysis more complete.

Section 2.2: change the verbs to past tense (e.g., the sample was removed, extract was vortexed, etc.).

Section 2.5.3: Since PeerJ is a very broad journal, add a few phrases that define/explain parameters like VIP, log2FC, etc.

Mention the mass spectrometry software, databases, spectral libraries, etc. used to identify the metabolites.

Specify how relative content of metabolites were calculated (was it peak area?) and if internal standards were used.

Specify the version numbers of R and R packages used.

Figures 2B and 7: in the figure captions, identify what parameter the colors refer to (e.g., log2FC).

In all figures, define error bars and statistical significance symbols (*, **, etc.).

·

Basic reporting

Title:
• Kindly include the specific LC-MS/MS (ESI-Q TRAP-MS/MS)

Authors and Affiliation:
• Are the authors' names correct? Please add a space to separate the first name and surname
• Include the zip code for the affiliation address

Abstract:
• The current version needs revision. I suggest that the authors revise the first two lines of the abstract, as the statements should be reserved for the introduction.
• Material composition can be ambiguous, and the authors should specify this.
• Specify the primary and secondary metabolites that were analyzed.
• In the results section of the abstract, please provide details on the differences in primary and secondary metabolites and indicate whether nectar or honey has higher content.
• Conclusion in the abstract is excellent and has powerfully conveyed the overall outcome of the study.

Keywords:
• Please include the scientific name of the loquat
• Loquat, as far as I know, is only produced in China, and it is suggested that the local name should be included.

Introduction:
• In the third paragraph, or lines 59-66, authors are advised to add more research to analyze metabolites and metabolite traceability in honey and nectar. For example, the study done by Islam et al. in 2024 (https://peerj.com/articles/achem-33/) and Sultana et al. in 2024 (https://www.mdpi.com/2304-8158/13/11/1739). With this, the research gap that the authors identified as a lack of complete analysis of metabolites will be valid since the said research only focused on a select secondary metabolites.
• In the third paragraph, or lines 65-66, it is suggested that the authors should specify that fully analyzing the metabolites pertains to the road-targeted metabolite profiling approach.
• Fourth paragraph, lines 71-73. A paper by Cheung et al identified p-hydroxybenzoic acid in loquat honey from China. Please include. https://www.tandfonline.com/doi/full/10.1080/10942912.2019.1579835?utm_source=chatgpt.com#abstract
• Fourth paragraph. Please add more health benefits of loquat honey, particularly in TCM, that are available.
• Fifth paragraph. Ensure that the aim and objective are written in the past tense.

Experimental design

Materials and methods.
• A separate section for the chemicals and reagents should be included.
• Section 2.1. Loquat nectar and honey samples. Based on this, honey production is uncontrolled, meaning bees can forage for nectar aside from the loquat. Sultana et al. 2024 (https://www.mdpi.com/2304-8158/13/11/1739) performed a controlled experiment from nectar to honey. Authors are requested to justify this and perhaps add statements regarding honey authentication since this factor can significantly influence honey composition.
• Section 2.1. Loquat nectar and honey samples. Please confirm that the honey collection is indeed in January 2023.
• Section 2.1. Loquat nectar and honey samples. Please provide more details on how the honey was processed. Did it undergo filtration? Also, indicate the storage conditions.
• Section 2.1. Loquat nectar and honey samples. Please specify the bee species that produced the honey.
• Section 2.2 and Section 2.3. These need revision to a more cohesive procedure.
• A separate section that details the compound matching should be provided, and the % match threshold should also be given.
• Section 2.2. Kindly specify the internal standard used and its concentration.
• Quantification was also performed, and the authors should provide details on how the compounds were quantified.
• Section 2.3 UPLC Conditions. Please provide information on the detector and the detection wavelength/s used.
• Question: Why was QTRAP used instead of QTOF?

Validity of the findings

Results:
• Figures. Ensure that DHPHM and DHPFM are defined in the figure description.
• Section 3.2.4 Saccharides. Doner 1977) is quite an obsolete reference. Please update this with a more recent resource. Furthermore, aside from glucose and fructose, maltose and sucrose are also considered the main saccharide components in honey.
• If the authors decide to have a separate discussion section, then it is advised that the results should only describe and narrate the results, and the result interpretation should be transferred to the discussion section.
• Authors should provide more details on how the quantified metabolites were chosen for quantification. What were the criteria? This should also be included in the methods section.
• Regarding the quantified compounds, authors must include the quantities in the results narrative.
• Figure 1 and Figure 7. Images are tiny, and it is advised that the authors separate A-D into individual compounds.
• Figure 2. Please explain the discrepancies in the heatmap of Lipids and Saccharides identified in the different replicates for the nectar samples.
• Figures 3 to 6. The t-test was used to compare statistical significance for these figures, particularly in C. Ensure that the criteria for p-value to determine significance are indicated in the methodology section. Furthermore, figure descriptions should also contain the meaning of *, **, ***, ****, etc.

Discussion:
• Authors should transfer the literature comparison in the results to this section.
• Authors should also discuss the differences among the flavonoids, saccharides, proteins, and phenolic acids. Also, explain the enzymatic process that brought the differences in glycosidic compounds.
• Authors should include literature that has attempted to compare the metabolomic profile of honey and nectar.
• Authors should include statements regarding the study's limitations, particularly concerning the study design and future directions.
• Authors should also include details that explain the significance of the study.

Additional comments

General Comments:
• Seems like there is a problem with spacing. Authors are advised to proofread their work for such errors.
• Ensure the English language is appropriate, especially for the methodology.

·

Basic reporting

Spelling check is required.
Line 13: Correct the word “abstract.”
Line 15: Change to “Honey bees produce honey.”
Lines 37-38: The sentence “Composed primarily of compounds” is redundant. It is suggested to change the word “compounds.”
Lines 42, 44, 61, 63: Review the writing and citation. Usually, when the author's name is written, the year is immediately included in parentheses. The quote on line 331 seems to me to be the correct way for all the previous ones.
Line 61: The study by Nicolson begins with a discussion, and at the end, two different authors are cited.
Line 74: The sentence is in the future tense. The study was already conducted and should be in the past tense.
Line 29: Include the meaning of the abbreviation OPLS-DA.

Line 192: Add a citation at the end of the sentence.
Line 264: There are sources. More recent bibliographical references change to a more current one
Line 327: Change the word "disparaties" to "differences"
331: Add citation
There is no discussion of saccharides

Experimental design

No comments

Validity of the findings

Figure 3, 4, 5, 6-C: What do *, **, and *** mean?
There is no discussion of saccharides, add.

Additional comments

The research is interesting, I think the discussion could be broader due to the large number of results.

---

## Round 0.2 · Minor Revisions

The authors have addressed my comments (Editor) along with Reviewer #2, as approrpriate. However, Reviewer #1 recommends improving the methods section. I also believe the methods section is ambiguous and it's difficult to understand fundamental information like how quantification was performed. Please address comments from Reviewer #1 by revising the methods. I suggest you refer to other articles published in your field that can give you ideas on how to better describe the methods you used.

**Language Note:** The review process has identified that the English language must be improved. PeerJ can provide language editing services - please contact us at [email protected] for pricing (be sure to provide your manuscript number and title). Alternatively, you should make your own arrangements to improve the language quality and provide details in your response letter. – PeerJ Staff

·

Basic reporting

Overall, current literature references had improved significantly as compared to the previous version and relevant articles suggested were added.

A lot of methodologies were not written properly and the authors are required to paraphrase the methods properly and accordingly to ensure cohesiveness. Also, ensure that English is correct, particularly in terms of grammar, punctuations, and spacing.

Article structure unfortunately needs improvement. Current version, particularly in the methodology part is disorganized, particulatly, in the case of sections 2.2 to 2.5. Section 2.2. on its own requires extensive rewording to ensure that the

Authors should carefully Asses the section and subsection title and ensure that it is a proper phrase that fully descripes the corresponding content of the sections or subsections. Also, authors are really requested to maintain consistency with the sentence cases of the titles of the sections and subsections.

Experimental design

Overall, research question and hypothesis is clear and research gap was carefully established.

Authors had now established the credibility and authenticity of loquat honey and nectar used. I however suggest that the pollen analysis should also be included as part of the manuscript, hence, the figure of pollen analysis be included in the main text (results section) and in the methodology section, the method in conducting the pollen analysis should be detailed and included. Also, in the pollen image, make sure that there should be two pollens, one for the honey and the other for the loquat pollen to establish the authenticity. This will irrevocably establish the identity of the sample.

Unfortunately, methods for the UPLC ESI Q-TOF requires more detailing, especially in the UPLC part. Please take note that this is the only experimental component of your paper and this should be detailed in such a way that the it can be reproduced by other researchers. I had observed that the UPLC methodologies had been deleted. Please ensure that it is still included in the manuscript.

In the response, authors had mentioned the chemicals and reagents used, however, table should be included in the manuscript to ensure transparency.

Quantification part of the experiment is still vague and unclear. Authors, can you please explain how did you performed the quantification? Were there subsequent phenolic and other compounds' standards used, or did you performed the quantification based on internal standard? This should be detailed carefully since bulk of your images/figures are based on quantification.

Validity of the findings

Results are mostly valid expect for the quantification part, which the authors need to address.

---

## Round 0.3 · accepted · Accept

All suggestions and comments were adequately addressed by the authors.

·

Basic reporting

During final proof reading, the publisher may suggest to rephrase those statements wherein first person voice was used (the use of we especially in the procedures).

Experimental design

The authors had addressed all my queries and had revised and added necessary information to address them.

Validity of the findings

MRM is a very powerful criteria which strengthens the validity of the quantification. Thanks you for including this information. Furthermore, the addition of a reference pollen in the image had improved the veracity of the samples used. Though, i wonder why the authors placed this very important document in the supplementary section.